# Influence of the Parameters of the Bus Lane and the Bus Stop on the Delays of Private and Public Transport

**Fadyushin Alexey** and **Zakharov Dmitrii** *

Department of Road Transport Operation, Industrial University of Tyumen, 625000 Tyumen, Russia; fadjushinaa@tyuiu.ru
* Correspondence: zaharovda@tyuiu.ru; Tel.: +7-908-874-6252

**Abstract:** The article deals with the influence of the infrastructure for public transport on the delay time of private and public transport in the city. The study employed the methods of simulation, mathematical modeling and field research. Imitation microscopic modeling determined the parameters of mathematical models of the delay time of private and public transport for various parameters of the bus lane, the length of the bus stop loading area, and its distance from the signalized intersection. Calculations determined the total delay time, taking into account the number of passengers in public and private transport on the section of the main street of regulated traffic. Determining the optimum parameters of the public transport infrastructure requires considering not only public transport passengers, but also drivers and passengers of private vehicles. Over-improving parameters of the bus lane has no effect on public transport, but traffic parameters for all other road users degrade. At high traffic intensity, the dependences of the total delay time on the length of the marking lines 1.11 and 1.5 are described by the parabola equation. The values for a road with three lanes have been determined, marking lines 1.11 and 1.5 at which the total delay time is minimal. For a highway with a high intensity, minimum bus stop parameters lead to significant increases in delay time.

**Keywords:** transport planning and modeling; urban transport system; public transport; private transport; bus lanes; bus stop; travel time; delay time

## 1. Introduction

It is hard to imagine a modern city without sustainable population mobility. Despite a significant decrease in population mobility in 2020 during the period of restrictions due to coronavirus/COVID-19 [1,2], experts predict a gradual recovery in transport demand. For a long time, the policy in the field of transport in Russian cities was of a catch-up nature and consisted of the construction of new highways, transport interchanges, tunnels, underground or elevated pedestrian crossings [3]. At the same time, the quality of transport services increased for users of private and public transport and deteriorated for pedestrians.

The trend of recent decades is urbanization, population growth in cities and their territories. In large cities of the world, sustainable mobility is achieved through the development of infrastructure and rolling stock of public transport, car and bicycle rental services, the creation of bike lanes and infrastructure for pedestrians [4,5]. To ensure sustainable urban mobility, priority conditions are created for public transport and cyclists in comparison with the use of a private car [6,7].

The authors in [8,9] show that in different cities the share of movements by different types of transport can differ significantly. At the same time, in most of the cities mentioned in [10],

residents prefer public transport. Both the development of public transport and ensuring the need for movement by public transport provide social justice for citizens [11,12].

When developing public transport in cities, it is important to form an efficient bus network that depends on transport demand [13]. The development of public transport, including its infrastructure, affects the economic performance of transport and the economy of the region [14,15]. The authors of [16] show that public transport affects the psychological state of people, and the development of infrastructure for public transport is one of the mechanisms for promoting health.

Some of the most important factors determining the choice of public transport by people for movement in cities is the travel time [17]. To reduce travel time, active and passive priorities are created for the movement of public transport [18–21]. To create an active priority for the movement of public transport when passing through signal intersections, three strategies are used: bus speed control, traffic signal control, or a combination of these [22]. Using an active priority reduces the travel time, the delay time, and the average queue length. Limiting the capacity of a stop affects the performance and efficiency of buses. The results of the study show the need for a comprehensive account of the active priority when crossing intersections for public transport and the throughput of bus stops. In addition, it is necessary to pay attention to ensuring the necessary throughput of intermediate and final bus stops [23–25]. Optimization of road infrastructure for public transport is an urgent scientific and practical task. Models of traffic on a road near a bus stop and methods of its placement were considered in the works [26–32]. For example, in [33], of the four types of bus stop considered, one of them includes the presence of an entrance pocket outside the traffic lane. When using stops of this type, the best road traffic parameters are achieved. However, the authors of [26–33] do not consider the effect of the length of the loading area on the traffic parameters with different transport demand.

In the paper [34], the influence of the location of the bus stop and distance from the intersection on the vehicle delay was established. With respect to the volume, the downstream bus stop is greatly superior to the upstream one when the distance is less than 70 m, and slightly inferior to the upstream one when the distance ranges from 70 m to 200 m. With regard to the vehicle delay, the upstream bus stop is better than the downstream one. However, the study uses constant values of traffic intensity and length of the loading area bus stop.

Much attention in the research is paid to the location of the bus stop on the route of public transport [35]. This is done to strike a balance between the speed of communication and the availability of the bus stop for pedestrians.

Increasing the average speed when creating bus lanes improves the operating conditions of electric buses which require stops to charge batteries with electric energy [36].

The creation of dedicated lanes for buses allows for organizing the movement of buses of different routes [37], which is important for large cities with a lack of highways and no off-street transport. However, even under such conditions, it is important to find the optimal parameters of the infrastructure for public transport which reduce the loss of time for people traveling by other modes of transport [38].

The aim of the work is to improve the efficiency of traffic management by establishing patterns of influence of the parameters of the infrastructure for public transport on the traffic delay time.

## 2. Materials and Methods

In this work, transport modeling at the microscopic level was used to assess the change in traffic parameters [39]. Microscopic modeling has become widespread in the optimization of traffic flows, design of road transport infrastructure facilities, optimization of traffic light facilities, simulation of the movement of unmanned vehicles as part of intelligent transport systems of cooperative control, taking into account fuel consumption of cars and emissions of harmful substances with car exhaust gases and when solving other problems [40–45]. Modeling allows you to simulate conditions that cannot be implemented in actual practice without significant financial and material inputs. Simulation modeling at the microscopic level requires calibrating the models used, taking into account real traffic conditions

and parameters [46]. To assess the effectiveness of traffic management, the delay time was selected. The work evaluated the average and total time delays in traffic.

The average delay time of the i-th mode of transport is the delay time for one vehicle of the i-th mode of transport moving along the studied section. Further in the work, the delay time of the i-th mode of transport is used.

Ratings of cities by the level of service traffic congestion commonly use a coefficient that characterizes how many times the travel time during peak hours $t_{trav}$ increases in comparison with free traffic $t_0$ [47]. The difference between $t_{trav}$ and $t_0$ will be the time delays $t_d$ created during the movement.

In the opinion of the authors of the article, it is not enough to take into account the parameters of traffic in relation to vehicles only. It is necessary to take into account the passenger capacity of vehicles and the number of passengers carried. On average, one passenger car accounts for 1.5 trips, and the simultaneous loading of buses can reach up to 100–120 passengers. Therefore, to take into account the time spent on movement, in this paper, the total delay time is used which is calculated by the equation:

$$T_d = \sum_{i=1}^{n} T_{di} = \frac{t_{di} \cdot Q_{di}}{3600} = \frac{t_{di} \cdot q_i \cdot N_i}{3600} \tag{1}$$

where $T_d$—total delay time per hour of academic time, h/1 h; $T_{di}$—total delay time of the i-th mode of transport, h; $t_{di}$—average delay time of the i-th mode of transport, s/person; $Q_i$—number of passengers carried by the i-th mode of transport on a section of the road network, persons/1h; $q_i$—average number of passengers in one vehicle of the i-th mode of transport, persons/vehicle; $N_i$—traffic intensity of the i-th type of transport, vehicle/1 h: $i$—serial number on the general list of modes of transport; $n$—the number of modes of transport on the general list.

The total delay time of the i-th mode of transport is the delay time of all users of the i-th mode of transport and drivers of private transport moving around the area under study.

The time spent on daily commuting to work and returning home can be a significant fraction of a day. Reducing this timing improves the quality of human life and the competitiveness of the city. In this work, the objective function has the form:

$$T_d \rightarrow min \tag{2}$$

where $T_d$—total delay time, h/1 h.

The object of research is a system shown in Figure 1.

Transport modeling was carried out in the PTV Vissim 11 software program. The strengths and weaknesses of the PTV Vissim software for modeling traffic flows are described in [48]. An important advantage is its good accuracy for busy traffic flows and large intersections.

In PTV Vissim, the simulation is performed with stochastic incoming flows, i.e., the time of vehicle arrival at the beginning of the segment is random. To improve the accuracy of the simulation results, 6 calculations of the average delay time were carried out during 10 min simulation time intervals and the average value of the delay time was calculated. These values are used in the article to check the adequacy of the mathematical models and the graphics.

The average delay time calculation starts at 1200 s of simulation and continues for 3600 s. In the period from 0 to 1200 s, the road network model is filled with vehicles. When level of service LoS > 1, at the moment of the start of determining the traffic parameters (from 1200 s), traffic jams are formed at the traffic lights in the simulation model. It gives an opportunity to achieve the correspondence of the parameters of the simulation model to real traffic conditions at a specific point in the simulation time.

The simulation model consists of a road, an approach to a regulated intersection and an intersection with traffic lights. The microscopic transport model of the city's traffic includes 1000 m long sections; traffic lights with a four-phase operation mode. Due to traffic light regulation, traffic flows at the intersection are separated in time, and each direction moves in a separate phase. The phase coefficient for the forward and right-hand traffic direction of the vehicle is 0.37, for the left-hand turning flow,

0.11. Modeling was carried out with the settings of the program corresponding to the optimal weather, climatic, and road conditions of vehicle operation. Simulation models were calibrated with respect to changes in maximum and desired acceleration and deceleration functions, the Wiedemann's car following model (look ahead distance, look back distance, average standstill distance, additive and multiplicative parts of safety distance), lane change and lateral (general behavior, active cooperative lane change, desired position at free flow), etc. Delay time was determined for a traffic flow moving in one direction.

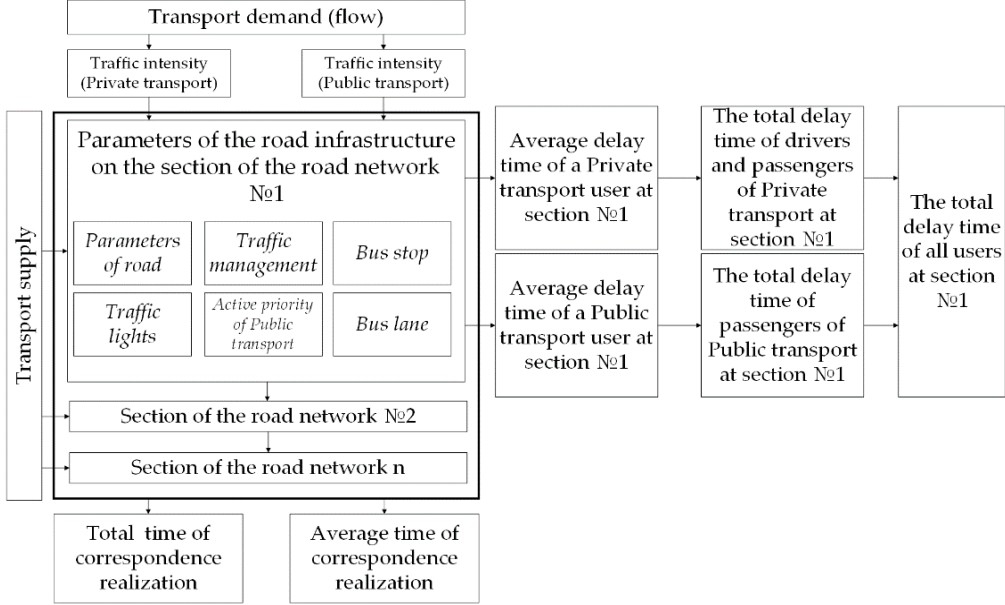

**Figure 1.** The structure of the system under study.

The experimental part of the study is carried out in several stages:

- Stage 1 involves the creation of simulation models of the movement of private and public transport in PTV Vissim 11 with parameters corresponding to the real conditions of movement of road transport on the road network;
- Stage 2 envisages field studies and determination of traffic flow parameters to calibrate traffic simulation models to match the actual traffic flow parameters;
- Stage 3 involves experimental research with the establishment of dependencies using simulation modeling;
- Stage 4 involves field research and measurement of the traffic delay time and the establishment of dependencies based on field research;
- Stage 5 provides for the analysis of research results and assessment of the adequacy of mathematical models to the results obtained in field research.

The article presents the intermediate results of the research on the first three stages using simulation modeling.

According to experts' forecasts, the number of residents in Tyumen will increase to 1 million by 2028–2029 and will reach 1.2 million by 2040. In 2016, in Tyumen, the average trip time for work purposes was 35 min. To maintain this parameter at a given level, the program for the integrated development of transport infrastructure plans to change the structure of population mobility and to increase the share of trips by public transport from 40 to 55% by 2040 (Figure 2).

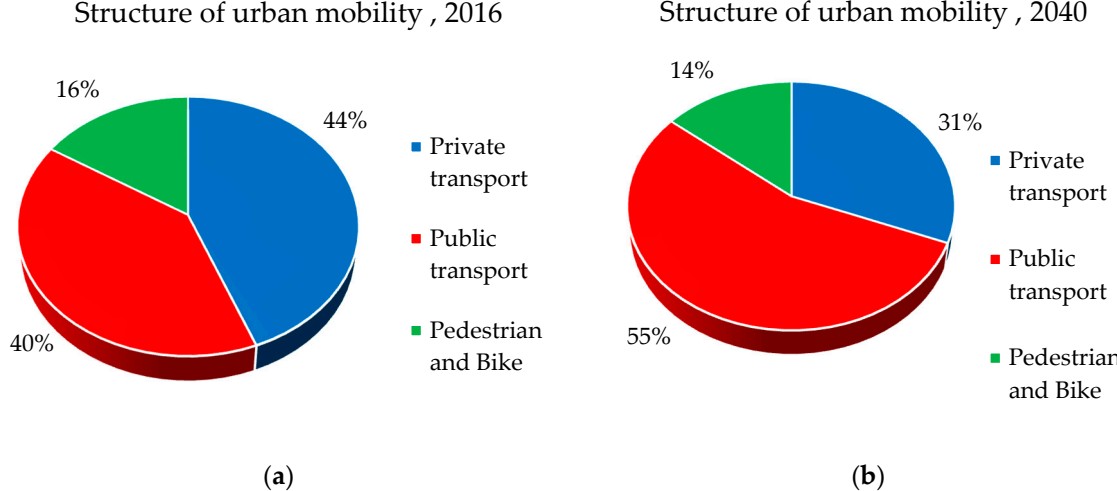

**Figure 2.** Structure of urban mobility according to the Program for the Integrated Development of Transport Infrastructure in Tyumen: (**a**) in 2016; (**b**) forecast for 2040.

Due to the expansion of the city's territory, it will be necessary to introduce new public transport routes, and to increase the number of buses. This requires an increase in funding from budgets of all levels. One way to improve the efficiency of public transport is to increase the average speed. This will reduce the need for rolling stock while maintaining headways or provide shorter headways without changing the number of buses. It is possible to increase the average traffic speed by creating active and passive traffic priorities for public transport. The passive priority of public transport traffic is the creation of bus lanes.

The greatest effect of their creation is achieved when creating a separate lane for buses. At the same time, the parameters of public transport traffic are improved and do not deteriorate for private transport. However, the built-up density and the limited right-of-way in most Russian cities allow for creating a bus lane only by reducing the number of lanes for private vehicles. At the same time, there will be an improvement in traffic conditions for public transport, but a deterioration in private ones. The lane load level for buses decreases, and for individual vehicles, increases.

The location of a bus stop relative to the signalized intersection can also affect the parameters of public and private transport. With high traffic intensity of public transport, the length of the bus stop loading area becomes significant. In the aggregate, the formation of optimal parameters of the infrastructure for public transport can reduce the time spent on travel.

As part of the transport planning documents, by 2040, the city plans for the organization of about 100 km of new bus lanes and 500 bus stops. If the bus lane parameters are selected incorrectly, a significant increase in the delay time on both private and public transport is possible. Conversely, by choosing the optimal bus lane parameters, it is possible to reduce the loss of time for all road users.

The work considered the following parameters of infrastructure for public transport:

- The length of the 1.11 continuous-dashed marking line before the signalized intersection;
- The length of the cancellation of the action of the bus lane (hereinafter—the 1.5 broken line marking) before the signalized intersection;
- The length of the bus stop loading area;
- The removal distance of the bus stop from the signalized intersection.

The range of the factor values of the variables under consideration includes:

- The length of the 1.11 marking line was varied from 20 to 60 m;
- The length of the 1.5 marking line was varied from 50 to 250 m;
- The lengths of loading area and removal distance of the bus stop were varied from 20 to 60 m;

- The traffic intensity of the private transport varied from 1000 to 2500 vehicle/h;
- The traffic intensity of the public transport varied from 60 to 240 bus/h;
- The average number of passengers in one bus varied from 20 to 80 persons/bus.

## 3. Results and Discussion

### 3.1. Influence of the Length of the 1.11 Bus Lane Marking Line on the Traffic Delay Time

When creating a bus lane on highways, it is usually placed in the far-right lane and separated from the second lane by a 1.1 solid marking line (Figure 3a). For the general motor vehicles to change to the far-right lane and make a right turn before the intersection, the 1.1 markings are replaced with 1.11. The bus lane marking parameters before the signalized intersection are regulated by the state standard GOST R 52289-2004 "Technical means of traffic management". The length of the 1.11 marking line before the intersection cannot exceed 80 m and ends 20 m or more before the intersection. In practice, the length of the 1.11 marking line may be 10 m. This situation is observed in the central parts of cities with a high density of the road network and a small distance between intersections, where a large number of public transport routes can pass.

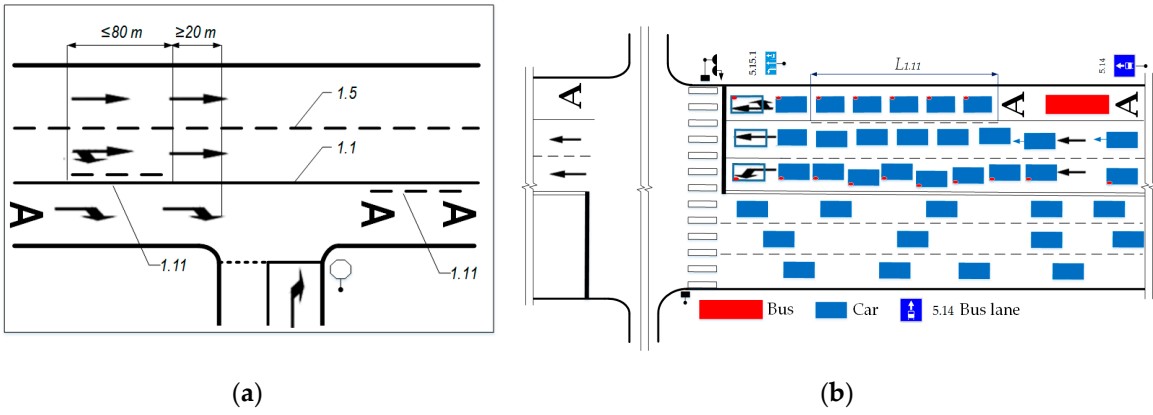

(**a**)　　　　　　　　　　　　　　　　　　　　　　　　　(**b**)

**Figure 3.** (**a**) Standardization of bus lane parameters before the intersection in accordance with GOST R 52289-2004. (**b**) Scheme of traffic on a road with a bus lane before a signalized intersection.

If the 1.11 marking line is long, a queue of 14–15 cars can form in the bus lane before the signalized intersection (Figure 3b). In this case, the bus may not have time to pass the intersection in one traffic light cycle, which will lead to an increase in the delay time.

The hypothesis of the study is that if the length of the 1.11 marking line before a signalized intersection increases, the delay time increases for public transport (hereinafter—PuT) and decreases for private transport (hereinafter—PrT).

The influence of the length of the 1.11 marking line before the signalized intersection on the average delay time of PrT and PuT is described by linear mathematical models (Equations (3) and (4)):

$$t_{dPrT} = t_{dPrT0} - S_L \cdot L_{1.11} \tag{3}$$

$$t_{dPuT} = t_{dPuT0} + S_L \cdot L_{1.11} \tag{4}$$

where $t_{dPrT}$—average PrT delay time, s; $t_{dPrT0}$—average PrT delay time at the minimum value length of the 1.11 marking line, s; $S_L$—parameter of sensitivity to changes in the length of the 1.11 marking line; $L_{1.11}$—length of the 1.11 marking line before the signalized intersection, m; $t_{dPuT}$—average PuT delay time, s; $t_{dPuT0}$—average PuT delay time at the minimum value length of the 1.11 marking line, s.

Figure 4 shows the graph of the dependence of the average delay time of PrT and PuT on the length of the 1.11 marking line before the signalized intersection.

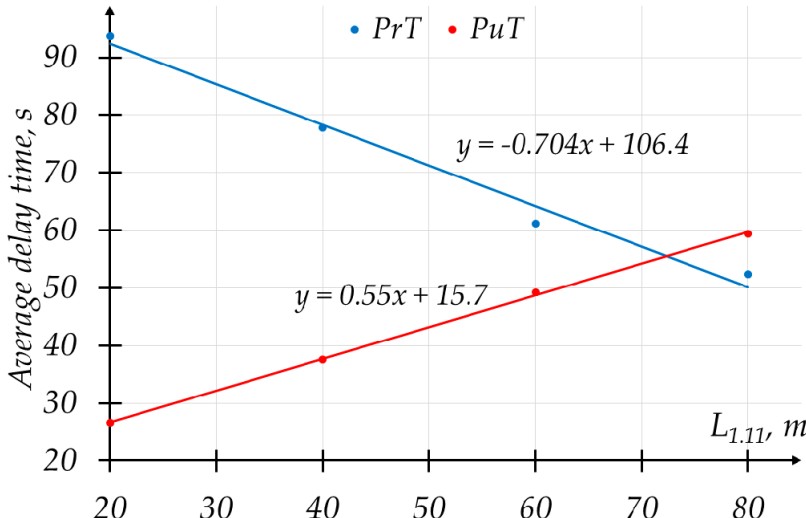

**Figure 4.** Influence length of the 1.11 marking line before a signalized intersection on a road with a bus lane on the average delay time of private transport (PrT) and public transport (PuT).

If the length of the 1.11 marking line increases from 20 to 80 m before a signalized intersection, the average delay time for PrT decreases by 47%, and for PuT it increases by 140%.

The study hypothesis is that with a high number of passengers on public transport, with an increase in length of the 1.11 marking line, the total delay time increases; with a low number, it decreases.

The linear one-factor model (Equation (5)) describes the effect of the length of the 1.11 marking line before the signalized intersection on the total PrT delay time. The model is one-factor because, on average, 1.5 passengers per trip travel in a private car. A multiplicative two-factor model that takes into account the length of the 1.11 marking line and the number of passengers in public transport (Equation (6)) describes the total delay time for PuT passengers.

$$T_{dPrT} = T_{dPrT0} - S_L \cdot L_{1.11} \tag{5}$$

$$T_{dPuT} = \begin{cases} (T_{dPuT0} + S_L \cdot L_{1.11}) \cdot q_{PuT} & \text{with } LoS < 1 \\ (T_{dPuT0} + S_L \cdot (L_{1.11} - L_0)^2) \cdot q_{PuT} & \text{with } LoS \geq 1 \end{cases} \tag{6}$$

where $T_{dPrT}$—total PrT delay time, h; $T_{dPrT0}$—total PrT delay time at the optimal value length of the 1.11 marking line at 1.5 trips in one vehicle, h; $T_{dPuT}$—total PuT delay time, h; $T_{dPuT0}$—total PuT delay time at the optimal value length of the 1.11 marking line, h; $q_{PuT}$—average number of passengers per bus, per./bus; $LoS$—level of service.

Based on the results of field observations, the following parameters are used in the simulation model: traffic intensity of PrT is 1500, 2000, 2500 veh./h; traffic intensity of PuT is 120 bus/h. In the work, the average number of passengers in one bus is one of the factors considered. For example, in Tyumen, in the central part of the city, buses of large or very large capacity operate (60% and 20%). The simulation model takes into account large-capacity buses. Figures 5 and 6 show the influence of the length of the 1.11 marking line before the signalized intersection and average number of passengers per bus on the total delay time.

By increasing the length 1.11 marking line from 20 to 80 m before the signalized intersection, the total PrT delay time is reduced by 50%, and the total PuT delay time increases in different ways, depending on the average number of passengers per bus. For example, with a loading of 20 per./bus, the total PuT delay time increased by 120%, and with 80 per./bus—128%.

With an average number of passengers, more than 40 per./bus, an increase in the length of the 1.11 marking line will lead to an increase in the total delay time (Figures 5b and 6). With 20 per./bus, an increase in the length of the 1.11 marking line will lead to a decrease in the total delay time.

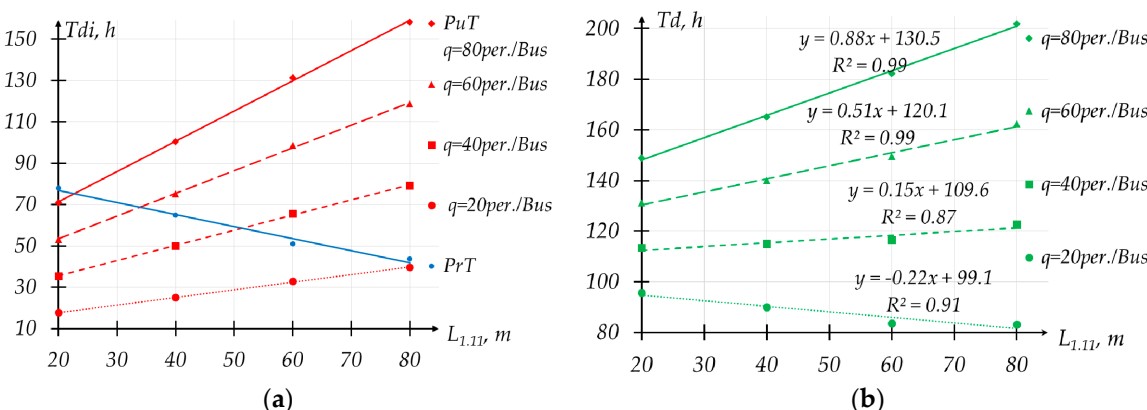

**Figure 5.** Influence of the length of the 1.11 marking line before a signalized intersection on a road with a bus lane on: (**a**) total delay time of PrT and PuT; (**b**) total delay time.

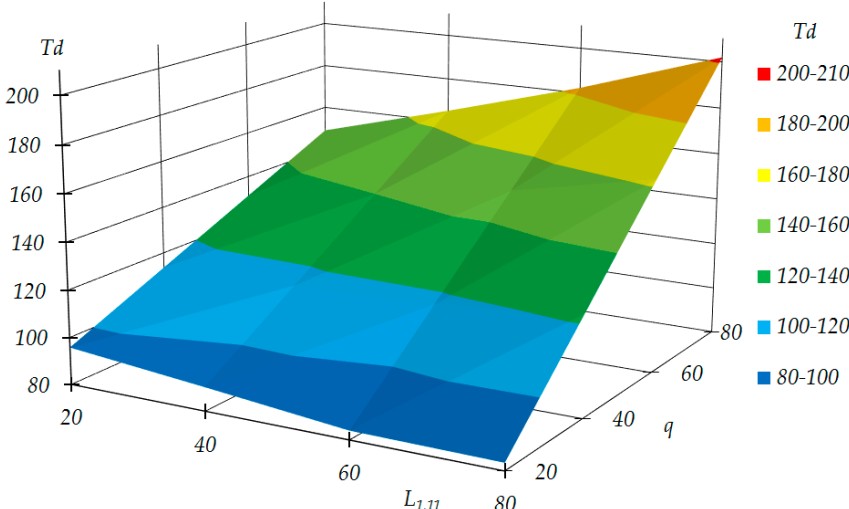

**Figure 6.** Influence of the length of the 1.11 marking line before a signalized intersection on a road with a bus lane and average number of passengers per bus on total delay time.

At low traffic intensity PrT, the change in the total delay time from the marking line 1.11 length is not significant. The influence of the marking line 1.11 length on the total delay time at different traffic intensities PrT is shown in Figure 7a. At 1500 veh./h, the length of the 1.11 marking line has practically no effect on the total delay time. At 2000 veh./h, with an increase in the marking line 1.11 length, a small increase in the total delay time is observed. The graph of the total delay time dependence on the marking line 1.11 length, built on the basis of empirical data, at LoS > 1 (or at 2500 veh./h) has the form of a parabola.

With a high traffic intensity PrT, the total delay time with an increase in the length of the marking line 1.11 for PrT decreases by 35%, for public transport it increases by 130%. Therefore, an extremum point appears where the total delay time is minimal, namely, 60 m (Figure 7b).

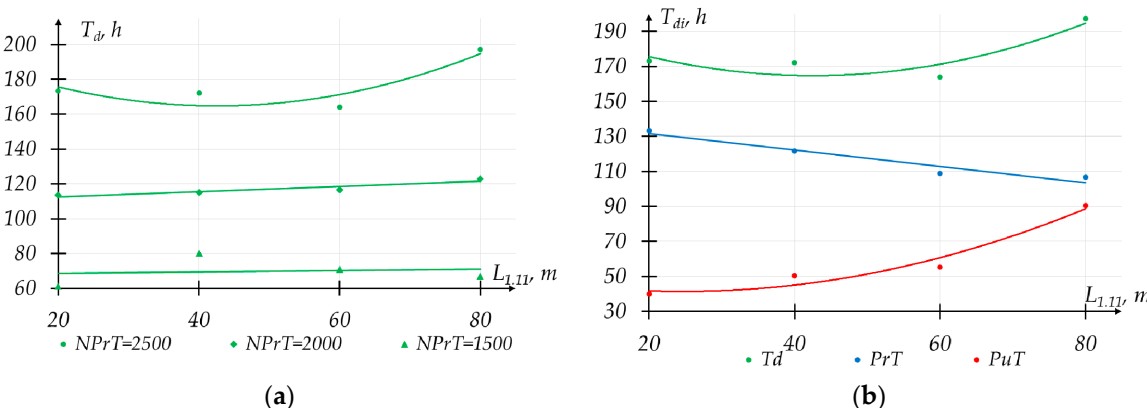

(**a**)                                         (**b**)

**Figure 7.** Influence of the length of the 1.11 marking line before a signalized intersection on a road with a bus lane on total delay time with average number of passengers 40 per./bus (**a**) with different traffic intensity PrT (**b**) with traffic intensity PrT 2500 veh./h.

Based on empirical data obtained by means of simulation modeling, the dependence of the total delay time from the marking line 1.11 length with traffic intensity equal to PrT 2500 veh./h and with average passenger number equal to 40 per./bus was developed. The dependencies presented in Figure 7b in mathematical form are presented in Equations (7)–(9).

$$T_{dPrT} = 0.464 - 140.7 \cdot L_{1.11} \tag{7}$$

$$T_{dPuT} = 41.39 + 0.015 \cdot (L_{1.11} - 24.5)^2 \tag{8}$$

$$T_d = 164.73 + 0.0214 \cdot (L_{1.11} - 42.56)^2 \tag{9}$$

### 3.2. Influence of the Length of the 1.5 Marking Line before a Signalized Intersection on the Traffic Delay Time

For the section of the main road of regulated traffic with a bus lane, there is a traffic organization scheme at an intersection with a left turn from the far-left lane (sign 5.15.1 "Lane traffic") and an additional traffic light section. With such a scheme, only one lane remains for the movement of PrT in the forward direction, which can lead to traffic congestion. To increase the capacity of the road, the bus lane ends at a certain distance before the signalized intersection and is marked with a road sign 5.14.1 "Bus lane end". The far-right lane is separated from the second lane by a 1.5 broken marking line, and in the signalized intersection area, a 1.7 line, with a shift in the trajectory of several lanes.

If the 1.5 marking line after the bus lane before the controlled intersection is long, a queue of PrT vehicles may form (Figure 8). In this case, the bus may not have time to pass the intersection in one traffic light cycle, which will lead to an increase in the delay time.

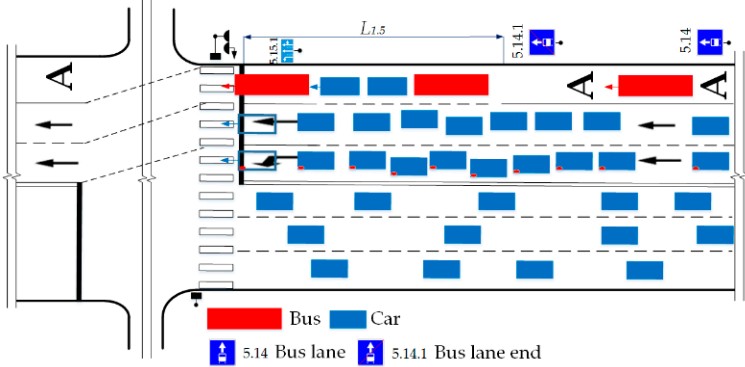

**Figure 8.** Scheme of traffic when the bus lane ends before a signalized intersection.

The research hypothesis is that if the distance between the stop line and the sign 5.14.1 "Bus lane end" (the length of the 1.5 marking line before the signalized intersection) increases, the delay time increases for PuT and decreases for PrT.

The influence of the length of the 1.5 marking line before a signalized intersection on the delay time of PrT and PuT is described by linear mathematical models (Equations (10) and (11)):

$$t_{dPrT} = t_{dPrT0} - S_L \cdot L_{1.5} \tag{10}$$

$$t_{dPuT} = t_{dPuT0} + S_L \cdot L_{1.5} \tag{11}$$

where $t_{dPrT}$—average PrT delay time, s; $t_{dPrT0}$—average PrT delay time at the minimum value length of the 1.5 marking line, s; $S_L$—parameter of sensitivity to changes in the length of the 1.5 marking line; $L_{1.5}$—length of the 1.5 marking line before the signalized intersection, m; $t_{dPuT}$—average PuT delay time, s; $t_{dPuT0}$—average PuT delay time at the minimum value length of the 1.5 marking line, s.

Figure 9 shows the graph of the dependence of the average delay time of PrT and PuT on the length of the 1.5 marking line before the signalized intersection.

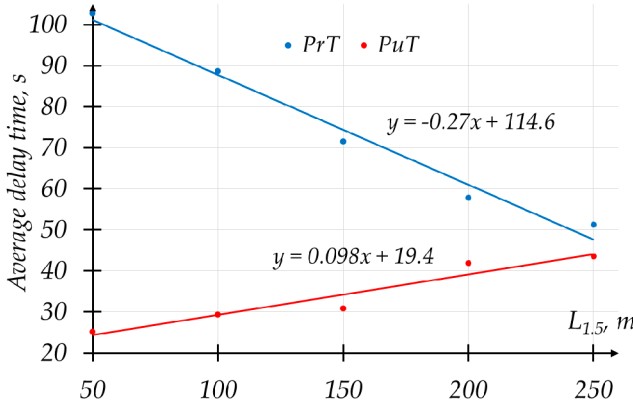

**Figure 9.** Influence length of the 1.5 marking line before a signalized intersection on a road with a bus lane on the average delay time.

If the length of the 1.5 marking line increases from 50 to 250 m before a signalized intersection, the average delay time for PrT decreases by 50%, and for PuT it increases by 80%. Such changes in delay time are more significant for PrT.

The study hypothesis is that with a high number of passengers on public transport, with an increase in length of the 1.5 marking line, the total delay time increases, with a low number—it decreases.

The influence of the length of the 1.5 marking line before a signalized intersection on the total delay time of PrT is described by a linear model (Equation (12)), and on the total delay time of PuT—by a multiplicative two-factor model (Equation (13)).

$$T_{dPrT} = T_{dPrT0} - S_L \cdot L_{1.5} \tag{12}$$

$$T_{dPuT} = \begin{cases} (T_{dPuT0} + S_L \cdot L_{1.5}) \cdot q_{PuT} & \text{with LoS} < 1 \\ (T_{dPuT0} + S_L \cdot (L_{1.5} - L_0)^2) \cdot q_{PuT} & \text{with LoS} \geq 1 \end{cases} \tag{13}$$

where $T_{dPrT}$—total PrT delay time, h; $T_{dPrT0}$—total PrT delay time at the optimal value length of the 1.5 marking line at 1.5 trips in one vehicle, h; $T_{dPuT}$—total PuT delay time, h; $T_{dPuT0}$—total PuT delay time at the optimal value length of the 1.5 marking line, h; $q_{PuT}$—average number of passengers per bus, per./bus; $L_0$—length of the 1.5 marking line before the signalized intersection at the minimum value $T_d$, m.

Figure 10 shows the influence of the length of the 1.5 marking line before the signalized intersection and average number of passengers per bus on the total delay time.

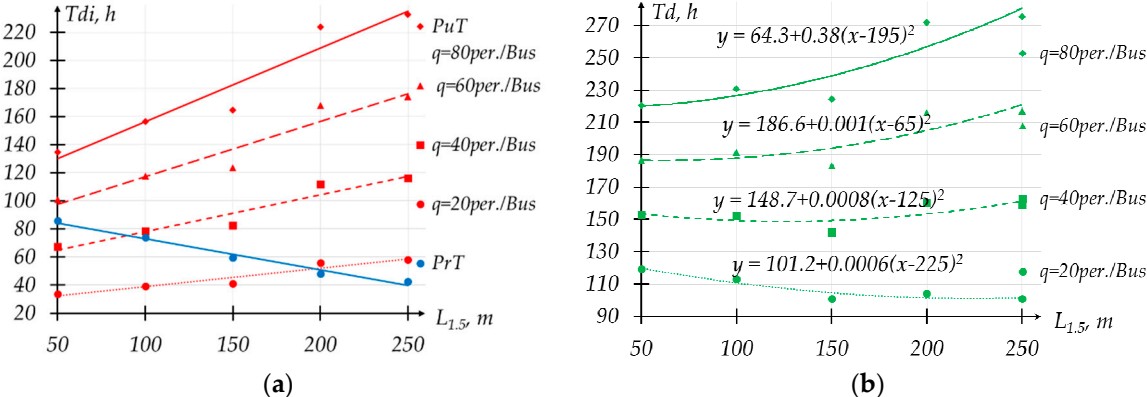

**Figure 10.** Influence of the length of the 1.5 marking line before a signalized intersection on: (**a**) total delay time of PrT and PuT; (**b**) total delay time.

If the length of the 1.5 marking line increases from 50 to 250 m before a signalized intersection, the average delay time for PrT decreases by 50%, and for PuT it increases in different ways, depending on the average number of passengers per bus. For example, with a load of 20 per./bus, the total PuT delay time increased by 100%, and with 80 per./bus—75%.

With an average number of passengers, more than 40 per./bus, an increase in the length of the 1.5 marking line will lead to an increase in the total delay time (Figure 10b). The influence of the length of the 1.5 marking line on the total delay time occurs in the form of a parabola. Consequently, there is a point with the optimal value length of the 1.5 marking line, where the total delay time is minimal.

With a high traffic intensity PrT (2500 veh./h), the total delay time with an increase in the length of the marking line 1.5 for PrT decreases by 26%, for PuT it increases by 130%. Therefore, an extremum point appears where the total delay time is minimal, namely, 150 m (Figure 11a).

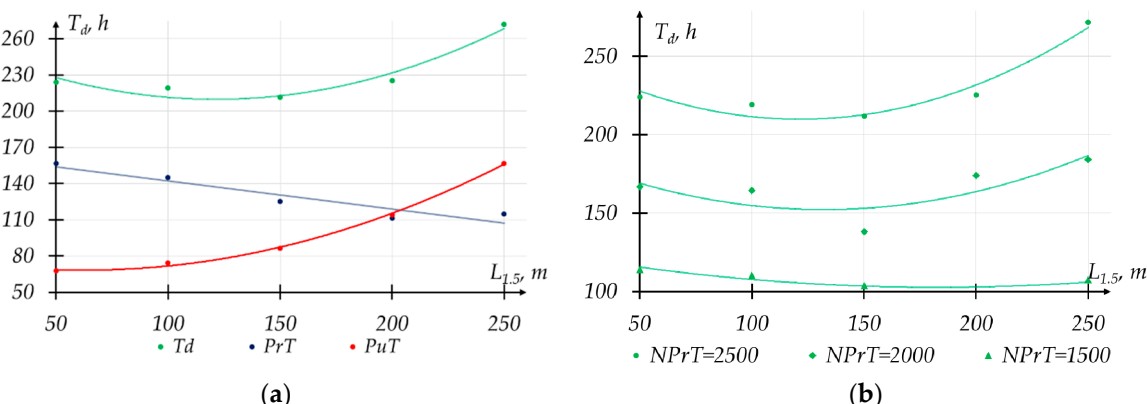

**Figure 11.** Influence of the length of the 1.5 marking line before a signalized intersection on total delay time with average number of passengers 40 per./bus (**a**) with traffic intensity PrT 2500 veh./h (**b**) with different traffic intensity PrT.

As the average number of passengers on public transport increases, the total delay time increases. However, at low traffic intensity PrT, the change in the total delay time from the length of the marking line 1.5 is less significant (Figure 11b). At 1500 veh./h, the length of the 1.5 marking line has practically no effect on the total delay time. The graph of the dependence of the total delay time on the length of

the marking line 1.5, built on the basis of empirical data, at LoS > 1 (or at 2000–2500 veh./h) has the form of a parabola.

At the minimum (50 m) and maximum (250 m) length of the marking line 1.5, the total delay time is higher (170 and 210 h, respectively) than at 150 m (150 h). Therefore, the length of the marking line 1.5 equal to 150 m is optimal in such conditions.

Based on empirical data obtained by means of simulation modeling, the dependence of the total delay time on the length of the marking line 1.5 with traffic intensity PrT 2500 veh./h and average number of passengers 40 per./bus was developed. The dependencies presented in Figure 11a in mathematical form are presented in Equations (14)–(16).

$$T_{dPrT} = 0.234 - 165.7 \cdot L_{1.5} \tag{14}$$

$$T_{dPuT} = 68.29 + 0.025 \cdot (L_{1.5} - 62.74)^2 \tag{15}$$

$$T_d = 209.87 + 0.035 \cdot (L_{1.5} - 121.38)^2 \tag{16}$$

### 3.3. Influence of the Bus Stop Parameters on the Delay Time

A bus stop for embarking and disembarking passengers is an important element of the PuT infrastructure. In Russia, general technical requirements for the elements of bus stops, the rules for their placement on highways, and their provision with technical means of traffic management are regulated by the industry standard OST 218.1.002-2003. The regulation allows for changing the length of the loading area in a range from 20 to 60 m. To make PuT accessible, the regulatory documentation recommends spacing stops 350–600 m between each other. Considering the district-based structure of the territory, the location of bus stops does not coincide with the recommendations for their location.

A bus stop can be placed before the signalized intersection under three conditions:

- There is a large passenger-forming point or an entrance to an underground pedestrian crossing before the intersection;
- The traffic capacity of the street before the intersection is greater than after the intersection;
- Immediately after the intersection, there is an approach to the transport engineering structure (bridge, tunnel, overpass) or there is a railway crossing.

Bus stops are usually located at the exit from the signalized intersection (after it), but with some exceptions, they can be located before the intersection. In both cases, with a high traffic intensity of PuT and PrT, traffic delays may occur (Figure 12). When creating new bus stops or reconstructing a road with the existing designers of the road facility, it is necessary to determine the optimal parameters.

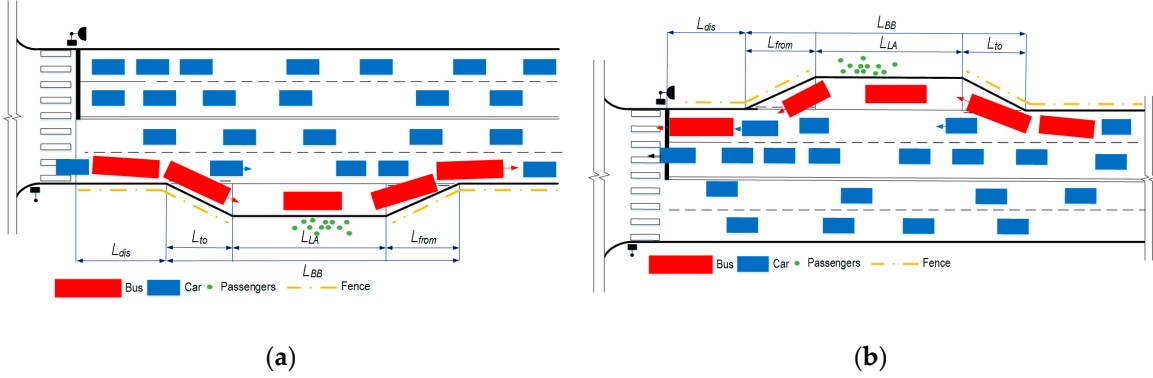

(**a**)                                        (**b**)

**Figure 12.** Scheme of traffic near the bus stop when it is located: (**a**) after the intersection; (**b**) before the intersection.

In a situation shown in Figure 12b, the bus stop is located before the signalized intersection, and the time delays are longer than when the bus stop is placed after the intersection. The paper considers the most common placing of a bus stop after a signalized intersection.

The research hypothesis is that if the length of the bus stop loading area increases, the delay time for PrT and PuT decreases; if the removal distance from the signalized intersection to the bus stop decreases, the delay time for PrT and PuT increases. The main delays for PrT and PuT in this situation consist of waiting for the arrival of buses at the bus stop, leaving the bus stop, and waiting for a permitting traffic signal. With an increase in the length of the bus stop loading area, the capacity the possible number of buses that can embark and disembark passengers increase. Thus, the wait time for a bus in the queue for entering the bus stop loading area decreases. The influence of the length of the loading area and the removal distance of the bus stop on the average delay time of PrT and PuT is described by linear, two-factor, additive mathematical models (Equations (17) and (18)):

$$t_{dPrT} = t_{dPrT0} - S_1 \cdot L_{dis} - a \cdot L_{LA}^b \tag{17}$$

$$t_{dPuT} = t_{dPuT0} - S_2 \cdot L_{dis} - c \cdot L_{LA}^d \tag{18}$$

where $t_{dPrT}$—average PrT delay time, s; $t_{dPrT0}$—average PrT delay time at the minimum value of the bus stop length, s; $S_L$—parameter of sensitivity to changes in variables; $L_{LA}$—length of the bus stop loading area, m; $L_{dis}$—removal distance of the bus stop from the signalized intersection, m; $t_{dPuT}$—average PuT delay time, s; $t_{dPuT0}$—average PuT delay time at the minimum value of the bus stop length, s; *a*, *b*, *c*, *d*—the model parameter of the power function.

Figure 13 shows graphs of dependences of the average delay time on the length of the bus stop loading area.

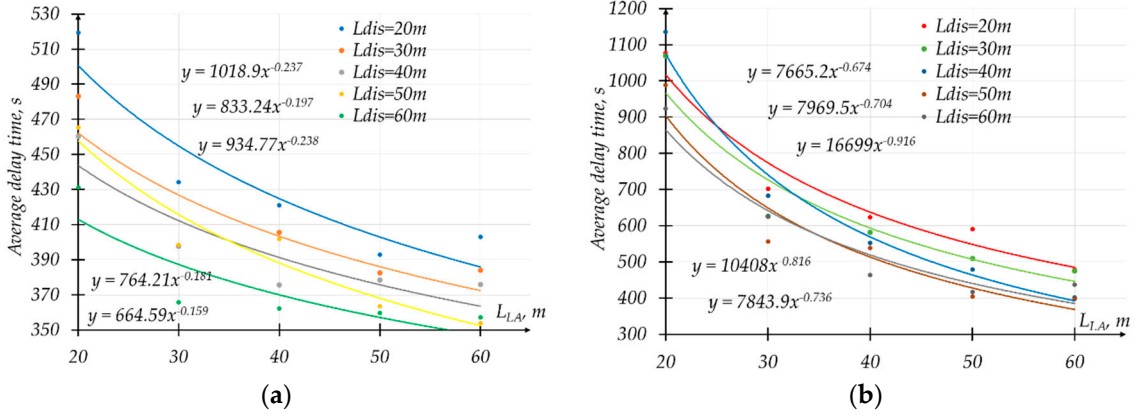

**Figure 13.** Influence of the length of the bus stop loading area on the average delay time of: (**a**) PrT; (**b**) PuT.

The influence of the length of the bus stop loading area on the average delay time of PrT and PuT (Figure 13) is described by a power function and depends on several factors: PuT traffic intensity, time spent on embarking and disembarking passengers, and PrT traffic intensity. If the length of the bus stop loading area increases, the average delay time for PrT and PuT decreases. This is due to a decrease in the queue of buses arriving at the bus stop.

Figure 14 shows graphs of the dependence of the average delay time on the removal distance of the bus stop.

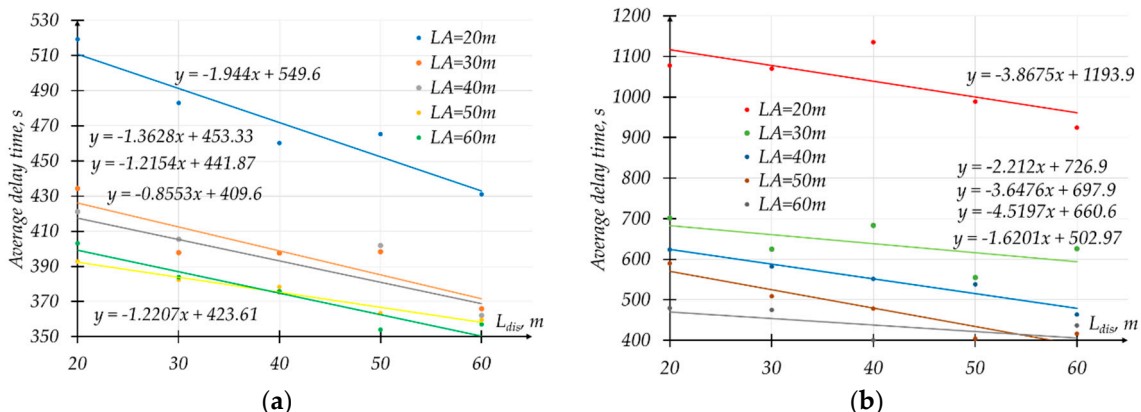

**Figure 14.** Influence of the removal distance of the bus stop on the average delay time of: (**a**) PrT; (**b**) PuT.

The influence of the removal distance of the bus stop from the signalized intersection on the average delay time of PrT and PuT (Figure 14) is linear and depends on several factors: PuT traffic intensity, throughput of the signalized intersection, time spent on embarking and disembarking passengers, and PrT traffic intensity.

A significant decrease in the delay time occurs with an increase in the loading area length to 30 m, and the removal distance to 30 m. A further increase in the length of the loading area has a less intensive effect. Consequently, there are optimal parameters of the bus stop, at which further changes in parameters will be insignificant and ineffective.

Since changing the parameters of the bus stop reduces the delay time for PrT and PuT, a comparison of changes in different modes of transport (by calculating the total delay time of the i-th mode of transport and total delay time) is not required.

The analysis of the research results showed that the parameters of the infrastructure for public transport affect the travel time, and, consequently, the efficiency of public transport. Taking into account the average time delay of PrT and PuT, all dependencies are linear, monotonically decreasing or increasing. It is not enough to draw a conclusion about the effectiveness of measures for the development of infrastructure for public transport by the values of the average delay time of one vehicle separately for PrT and PuT. It is necessary to take into account the loss of time of all users of these transport systems. In this case, the minimum loss of time is achieved with other parameters of the infrastructure for PuT.

The novelty of the research results lies in the establishment of new dependencies:

- Delay time on the length of the marking 1.11 before the regulated intersection on the road with the bus lane;
- Delay time on the length of the marking 1.5 before the regulated intersection on the road with the bus lane;
- Delay time on the length of bus stop loading area;
- Delay time on the removal distance of the bus stop from the regulated intersection.

The results and dependencies obtained are universal and can be applied to other city transport systems, for the following reasons:

- Technical characteristics of vehicles (car, bus, truck) operated in the USA, EC, Russia are approximately at the same level;
- The road rules adopted in different world countries are often alike, since these countries (including Russia) have joined the Vienna Convention on Road Traffic;
- Traffic organization schemes (including Bus lane and Bus stop) are approximately the same in different countries;

- The versatility of software products for traffic modeling;
- The problem of traffic jams and improving the service level is typical for large cities in all world countries.

The established dependencies will specify measures to optimize the parameters of the public transport infrastructure (length of the marking 1.11 or 1.5 before the regulated intersection on the road with the bus lane, bus stop parameters). This allows municipal authorities and urban transport planners to improve the quality of passenger traffic and, in some cases, the efficiency of traffic management. Additionally, the following effects are obtained:

- Reducing the number of citizens' complaints about the work of public transport and increasing the satisfaction of citizens with the work of municipal authorities;
- The travel time for urban residents will decrease;
- The efficiency of highways will increase due to a more even loading of the road lanes and an increase in the service life of the asphalt concrete pavement on the lanes for private vehicles;
- The level of aggression and the number of conflicts on the part of drivers of private transport will decrease by reducing the idle time at intersections;
- The number of emergency situations will decrease due to a more even movement of traffic flows and a decrease in the number of line changes on a limited section of the road;
- Emissions of pollutants with car exhaust gases will decrease.

The introduction of separate lanes for public transport will make it possible to more effectively use the priority of crossing intersections for public transport through adaptive control of traffic lights.

## 4. Conclusions

The results show that when LoS < 1, optimization of the parameters of the bus lane and bus stop is not required; however, the largest number of passengers, in cars and buses, is on the road network during peak hours. Based on the simulation results, at high traffic intensity (LoS ≥ 1), corresponding to peak hours, the influence of the parameters of the infrastructure for public transport on the delay time is not linear and it is possible to determine the optimal value's length of the marking line 1.11 or 1.5.

In the paper, the proposed mathematical model is derived from using simulation modeling. This model has some limitations. We assume that bus-overtaking maneuvers are prohibited and that the probability that the bus on each berth first finishes passenger serving is identical when all the berths are occupied. In addition, we assume that passengers board and disembark at a fixed time.

Transport modeling makes it possible to determine the optimal parameters of the infrastructure for PuT on the road according to the criterion of traffic management efficiency. These parameters must be taken into account in the design of new and reconstruction of existing highways. Unfortunately, at present, insufficient attention is paid to the infrastructure for public transport in the design of roads. The country's authorities began to pay increased attention to the development of public transport in Russian cities. Programs for renewing the rolling stock of fixed-route transport in cities and co-financing of costs from the federal budget have appeared. However, the purchase of new rolling stock alone cannot solve the goals set in the urban transport planning documents. A set of measures is required to reduce the proportion of movements by private transport and improve the quality of public transport.

Traffic simulation is very widespread in the world. Its application allows you to "look into the future" and assess transport demand and road conditions in a situation that is not currently observed. According to some European studies, in order to maintain a social distance to reduce the risk of COVID-19 disease, the carrying capacity of public transport while maintaining passenger traffic that sufficed before the pandemic should increase by 2.5 times. The authors of the article plan to continue research on the topic and conduct field tests in order to assess the compliance of the research results with the actual parameters of the total delay time. The existing road and transport infrastructure in large cities of Russia was created according to outdated regulatory documents and often does not

meet new challenges, such as the growth of motorization, the introduction in cities of measures to prioritize public transport, and the impact of the COVID-19 pandemic on society, the economy and the transport system cities. Therefore, according to the authors, to update the regulatory documents requires research on this topic.

**Author Contributions:** Conceptualization, Z.D.; formal analysis, F.A.; investigation, F.A.; project administration, F.A.; methodology, F.A.; supervision, Z.D.; validation, Z.D.; data curation, F.A.; funding acquisition, Z.D.; writing—review and editing, F.A., Z.D. All authors have read and agreed to the published version of the manuscript.

**Funding:** The article was prepared as part of the implementation of a state assignment in the field of science for scientific projects carried out by teams of researchers in scientific laboratories of higher educational institutions subordinate to the Russian Ministry of Education and Science on the project: "New patterns and solutions for the functioning of urban transport systems in the paradigm 'Transition from owning a personal car to mobility as a service'" (No. 0825-2020-0014, 2020–2022).

**Conflicts of Interest:** The authors declare no conflict of the interest.

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
