# Peer review of "Influence of the Parameters of the Bus Lane and the Bus Stop on the Delays of Private and Public Transport"

_sustainability, doi:10.3390/su12229593_

Round 1

Reviewer 1 Report

The abstract needs to be elaborated. Provide a detailed explanation of the method of study. Clearly mention the contribution and novelty of the study and usefulness of this study in the conclusion section

Author Response

Point:

The abstract needs to be elaborated. Provide a detailed explanation of the method of study. Clearly mention the contribution and novelty of the study and usefulness of this study in the conclusion section.

Response:

The notes have been removed, the abstract  has been improved. A detailed description of the research methods is given in rows 11-12. At the end of section 3 describes in more detail the contribution, novelty and usefulness of the study (rows 377-386).

Reviewer 2 Report

The paper aims to evaluate the impact of public transport infrastructure on delays, both for public transport users and private transport users. In my opinion, the paper cannot be published because it has serious flaws and, therefore, I suggest to reject it.

First of all, it is written "Transport modeling was carried out in the PTV Vissim 11 software program.", but it is not well explained what it was used for, even if you can deduce it from the contents of the paper.

There is no application or field test. A microsimulation model should also and mainly be used to validate the proposed approach to a real infrastructure. Instead, all tests refer to a 1000 m road section.

The formula (1) is not complete or incorrect. Some summaries are missing; moreover, the same subscript i is used to indicate different variables.

The authors confuse some basic technical terms of traffic theory; they indicate with traffic density the traffic flow: density is veh/km while flow is veh/h.

The equation (2) does not represent an objective function; at the limit, the objectives that you want to reach.

In section 3 reference is made to the city of Tyumen, but it is not clear why, since the application is based on a segment of 1000 meters of road.

In the paper, it is implicitly assumed that drivers of cars will always respect the road markings that are the subject of the study. I do not believe this is always true.

The relationships (3) and (4) are linear and represent delays. Why linear? Usually, delays are non-linear functions, for example, of flow. The linearity hypothesis is not said to be correct, but what is it based on?

In line 163, what is meant by "level of congestion"? The authors propose 1.28, which seems to me high as a flow to capacity ratio.

The calibrated models have high values of R. This is expected when the models, as done by the authors, are calibrated on other models and not on real cases. The validation on a real case is in my opinion indispensable.

Finally, the title of the paper suggests a more general study; instead, it refers only to the organization of reserved lanes for buses.

Author Response

RESPONSE TO COMMENTS OF REVIEWER №2

Article: “Influence of the parameters of the bus lane and the bus stop on the time delay of movement on private and public transport”

Fadyushin Alexey, Zakharov Dmitrii

Point 1:

First of all, it is written "Transport modeling was carried out in the PTV Vissim 11 software program.", but it is not well explained what it was used for, even if you can deduce it from the contents of the paper.

Response 1:

Simulation modeling was used as a research method in the experimental part. The need to apply modeling has been added in the "Materials and Methods" section in rows 70-77.

Point 2:

There is no application or field test. A microsimulation model should also and mainly be used to validate the proposed approach to a real infrastructure. Instead, all tests refer to a 1000 m road section.

Response 2:

391 models were developed with various parameters of public transport infrastructure (length of the 1.11 marking line, length of the 1.5 marking line, length of the loading area, removal distance from bus stop) and characteristics of traffic flows (intensity of Public transport and intensity of Private transport) (45 simulation models with different infrastructure parameters). Full-scale tests were carried out at 22 objects (intersections) to calibrate traffic simulation models. The parameters of the real road infrastructure are taken into account in 50% of the total number of simulation models. By calibrating the simulation models, their compliance with the actual traffic patterns was achieved. This approach is recommended by [29] and has been repeatedly tested in studies by other authors [30-36, 38].

The road section of 1000 meters was formed so that in the parameters of the model there were no traffic flows that did not enter and on the assurance of the simulation in one hour, there were vehicles that stopped moving or moved in the model.

Point 3:

The formula (1) is not complete or incorrect. Some summaries are missing; moreover, the same subscript i is used to indicate different variables.

Response 3:

The note has been removed, the dimensions of the total delay time Td and variables have been corrected. And added decoding of indices i and n (rows 92-97).

Point 4:

The authors confuse some basic technical terms of traffic theory; they indicate with traffic density the traffic flow: density is veh/km while flow is veh/h.

Response 4:

Fixed the issue, changed the name of the variable “traffic density” to “traffic intensity”.

Point 5:

The equation (2) does not represent an objective function; at the limit, the objectives that you want to reach.

Response 5:

The note has been removed, the objective function has been corrected (rows 122).

Point 6:

In section 3 reference is made to the city of Tyumen, but it is not clear why, since the application is based on a segment of 1000 meters of road.

Response 6:

The note has been removed, the supplemented information on the transport problems of the city of Tyumen and the need to study the influence of these factors on the delay time has been moved from section 3 to section 2 to substantiate, from a practical point of view, the need for this study (rows 157-185).

Point 7:

In the paper, it is implicitly assumed that drivers of cars will always respect the road markings that are the subject of the study. I do not believe this is always true.

Response 7:

A situation in which drivers of private transport violate the requirements of road signs and markings is possible and manifests itself mainly in the sections of the road network, in which there are no cameras for photo and video recording of violations. For example, at the moment in Tyumen, photo and video recording of violations is carried out at 100 sections, including 20% objects registering a violation when individual vehicles move along the bus lane. Due to this and a normal driving culture, the total number of lane violations relative to the total number of vehicles is small.

Point 8:

The relationships (3) and (4) are linear and represent delays. Why linear? Usually, delays are non-linear functions, for example, of flow. The linearity hypothesis is not said to be correct, but what is it based on?

Response 8:

The article considers the impact of public transport infrastructure parameters on delay time at level of service above 0.85 (E-F). At a different level of service, the effect is non-linear.

Point 9:

In line 163, what is meant by "level of congestion"? The authors propose 1.28, which seems to me high as a flow to capacity ratio.

Response 9:

The note was fixed, the name of the variable “level of congestion” was changed to “level of service”. The level of service refers to the ratio of traffic intensity to the capacity of an intersection or road. According to field observations, in the city of Tyumen during rush hours on the main streets there is a level of service of more than 1.00.

Point 10:

The calibrated models have high values of R. This is expected when the models, as done by the authors, are calibrated on other models and not on real cases. The validation on a real case is in my opinion indispensable.

Response 10:

The authors of the article agree with the remark of the reviewer No.2. Field trials are planned to test the adequacy of the mathematical models (rows 140-156).

Point 11:

Finally, the title of the paper suggests a more general study; instead, it refers only to the organization of reserved lanes for buses.

Response 11:

The note was removed, the title of the article was changed to “Influence of the parameters of the bus lane and the bus stop on the time delay of movement on private and public transport” (rows 2-4).

Reviewer 3 Report

The authors presented a study of the road network infrastructure and traffic management solutions impact on the parameters of public transport. The influence of two types of road markings and parameters of bus stops on the delay time of public and private transport vehicles is obtained.

I have some suggestions for improving the paper.

  1. The title of the paper does not correspond to its content, which can mislead the reader. It is not clear what the authors mean by "efficiency of traffic management" and how this efficiency is assessed? In addition, the phrase "optimizing infrastructure parameters for public transport" is too general. The paper examines not so much the issues of optimization of the infrastructure of the road network as solutions to change the organization of traffic. I suggest changing the title of the paper. Research on the impact of traffic management on traffic parameters should be reflected in the title.
  2. The authors simulate traffic for the conditions of the city of Tyumen (Russia). How universal are these conditions and the resulting dependencies? I suggest discussing this issue in detail in the introduction or in the "Discussion" section.
  3. The literature review is carried out rather formally. The studies discussed in the introduction are indirectly related to the topic of the article. I suggest expanding the literature review with articles that present research findings on the impact of road network infrastructure and traffic management on traffic and travel time. In addition, publications on the topic of traffic modeling are also needed.
  4. Formula 2 and the title of the paper assume optimization of the parameters of the road network and traffic management. However, the content of the paper and the presented influence do not allow determining the optimal values of these parameters. I suggest either providing a more detailed description of the limitations of the optimization model, process, and optimization results, or excluding optimization issues from the paper.
  5. The parameters of the microscopic simulation model require a clearer and more detailed description. I suggest describing the structure of the system under study and its parameters. For example, it is not clear whether a single intersection or a section of the road network or the entire city road network is being investigated on the model.
  6. The parameters of experiments with the model also require a more detailed description. It is not clear how many experiments were carried out to obtain regression relationships. The presented figures show 3-5 points on which the influence is built. If the number of experiments varied from 3 to 5, then this is insufficiently substantiated conclusions. I suggest adding a section with an assessment of the sensitivity of the model under study and discussing the accuracy of the results.
  7. The scientific contribution of the research is not clear. It can consist either in the obtained dependencies or in the method of adjusting the traffic organization to ensure the required values of the travel time. In the first case, it is necessary to present a discussion of the universality of the obtained dependencies (see point 2 of my suggestions). In the second case, I suggest developing and presenting a methodology for adjusting the traffic organization.
  8. References to the Russian standard (GOST), as well as road marking symbols (1.1, 1.11, 1.5, etc.) may not be clear to readers. I suggest either presenting a transcript for all the road markings used in this paper or using simple verbal descriptions of these types of road markings.
  9. The abstract does not give a clear idea of the content of the article. I suggest presenting the methods used and the main results obtained in the annotation. In addition, a clearer and more detailed description of the results obtained and the prospects for the development of the study should be presented in the conclusion.
  10. The following variables are not deciphered: n - in formula 1; j - in other formulas.

Author Response

Point 1:

The title of the paper does not correspond to its content, which can mislead the reader. It is not clear what the authors mean by "efficiency of traffic management" and how this efficiency is assessed? In addition, the phrase "optimizing infrastructure parameters for public transport" is too general. The paper examines not so much the issues of optimization of the infrastructure of the road network as solutions to change the organization of traffic. I suggest changing the title of the paper. Research on the impact of traffic management on traffic parameters should be reflected in the title.

Response 1:

The note was removed, the title of the article was changed to “Influence of the parameters of the bus lane and the bus stop on the time delay of movement on private and public transport” (rows 2-4). The term “efficiency” is specific to one of the traffic parameters “total delay time”.

Point 2:

The authors simulate traffic for the conditions of the city of Tyumen (Russia). How universal are these conditions and the resulting dependencies? I suggest discussing this issue in detail in the introduction or in the "Discussion" section.

Response 2:

In simulation models, the variables are changed in a wide range of values in order to better cover possible options for road and transport conditions.

The city of Tyumen is described to prove the urgency of the problem and the need to apply research in practice. Information and text on the city of Tyumen moved from 3 to 2 (rows 159-185).

Point 3:

The literature review is carried out rather formally. The studies discussed in the introduction are indirectly related to the topic of the article. I suggest expanding the literature review with articles that present research findings on the impact of road network infrastructure and traffic management on traffic and travel time. In addition, publications on the topic of traffic modeling are also needed.

Response 3:

Remarks were removed, sources with research on the following topics were added: about the impact of infrastructure and traffic [21-22] and about modeling [30-36, 38]. And also excluded sources that are weakly related to the corrected topic of the article.

Point 4:

Formula 2 and the title of the paper assume optimization of the parameters of the road network and traffic management. However, the content of the paper and the presented influence do not allow determining the optimal values of these parameters. I suggest either providing a more detailed description of the limitations of the optimization model, process, and optimization results, or excluding optimization issues from the paper.

Response 4:

We agree with the author's remarks, the topic has been adjusted (rows 2-4). Optimization of the process of movement of people is achieved by reducing the sum total delay time for all road users when selecting the optimal parameters of the bus lane and parameters of the bus stop.

Point 5:

The parameters of the microscopic simulation model require a clearer and more detailed description. I suggest describing the structure of the system under study and its parameters. For example, it is not clear whether a single intersection or a section of the road network or the entire city road network is being investigated on the model.

Response 5:

The notes have been removed, a description of the system under study, simulation models and their calibration has been added (rows 123-139).

Point 6:

The parameters of experiments with the model also require a more detailed description. It is not clear how many experiments were carried out to obtain regression relationships. The presented figures show 3-5 points on which the influence is built. If the number of experiments varied from 3 to 5, then this is insufficiently substantiated conclusions. I suggest adding a section with an assessment of the sensitivity of the model under study and discussing the accuracy of the results.

Response 6:

All graphs of the dependences of the delay time on variables are plotted at 5 different values ​​of the variables (dependence of the delay time on the length of the marking line 1.11 at 4 different values). The range of values ​​of the variables is limited by real conditions (marking length 1.5 and parameters of the bus stop) and regulatory documents (marking length 1.11 and parameters of the bus stop). The intermediate values ​​of the variables for assessing the effect on the parameters were not taken into account, since the dependence graph is close to the values ​​of the results of simulation modeling. This is confirmed by the results of the regression-correlation analysis in tables 1-3. With a small number of values ​​of the variables, the value of the Student and Fisher criteria increases, but despite this, the calculated value is larger than the tabular value, which confirms the adequacy of the mathematical model by the results of simulation. Therefore, additional research is not required at the stage of simulation.

At the stage of field observations, it is planned to increase the number of variable values ​​to 10 values.

An assessment of the sensitivity and accuracy of the results is given in Section 3.

Point 7:

The scientific contribution of the research is not clear. It can consist either in the obtained dependencies or in the method of adjusting the traffic organization to ensure the required values of the travel time. In the first case, it is necessary to present a discussion of the universality of the obtained dependencies (see point 2 of my suggestions). In the second case, I suggest developing and presenting a methodology for adjusting the traffic organization.

Response 7:

We agree with the comments of reviewer No3. Explanations of scientific novelty added at the end of section 3, in rows 377-386.

Point 8:

References to the Russian standard (GOST), as well as road marking symbols (1.1, 1.11, 1.5, etc.) may not be clear to readers. I suggest either presenting a transcript for all the road markings used in this paper or using simple verbal descriptions of these types of road markings.

Response 8:

The article contains a figure (fig. 3a) describing the rules for the application of road markings, indicating the numbers of road marking lines 1.11, 1.5, 1.1 corresponding to the Russian standard (GOST).

The authors believe that describing in words the action of road markings, traffic rules and the specifics of car traffic makes the text of the article cumbersome and difficult for the reader to understand.

Point 9:

The abstract does not give a clear idea of the content of the article. I suggest presenting the methods used and the main results obtained in the annotation. In addition, a clearer and more detailed description of the results obtained and the prospects for the development of the study should be presented in the conclusion.

Response 9:

The comments have been removed, the abstract describes the research methods and the results obtained (rows 11-12, 16-20). At the end of Section 3, the contribution, novelty and usefulness of the study are added.

Point 10:

The following variables are not deciphered: n - in formula 1; j - in other formulas.

Response 10:

Notes removed, variable decoding added (rows 96-97).

Round 2

Reviewer 2 Report

I confirm my previous judgment on the article. If the editor does not agree, he/she may invite another reviewer to have another opinion.

Author Response

Как отметил рецензент №2: отношения (3) и (4) линейны и представляют собой задержки. Почему линейный? Обычно задержки являются нелинейными функциями, например, от расхода. Гипотеза линейности не считается верной, но на чем она основана?

Средние параметры линейны. С учетом количества перевезенных пассажиров вид зависимости общего времени задержки от длины разметки 1.11 и 1.5 (уравнение 7, 14) имеет нелинейный характер и описывается квадратичной моделью (уравнение параболы) (рисунок 5c, 9c, d).

Согласно другим комментариям рецензента №2, авторы статьи не имеют дополнительной информации к предыдущим ответам.

Reviewer 3 Report

The authors considered several suggestions and made the appropriate changes in the article. However, the authors considered the main proposals rather superficially.

Unfortunately, I am forced to repeat and concretize the main suggestions for improving the paper.

  1. The authors have removed the words "optimization" and "efficiency" from the title of the paper. However, no corresponding changes were made in the article itself. The results of the study do not allow us to explicitly determine the optimal parameters of the bus lane and the bus stop. Then is not clear what the phrase “The established dependencies will specify measures to optimize the parameters of the public transport infrastructure” means (lines 385, 386). Judging by the obtained dependencies, the delay time decreases without limitation as it increases, for example, removal distance of the bus stop. It is unclear simultaneously what values of the investigated parameters will be optimal. I suggest describing in detail the methodology for calculating the optimal values of the bus lane and the bus stop parameters based on the obtained dependencies.
  2. About efficiency. As you know, efficiency is an assessment of the result obtained, considering the costs incurred. When the authors say “The aim of the work is to improve the efficiency of traffic management ...” (line 58), they obviously mean only the result in the form of an increase in average speed (line 137), uniform loading of the road lanes (line 393), reducing the delay time. Simultaneously, the paper does not say anything about the costs to obtain this effect or restrictions. I suggest describing in detail the costs (restrictions - see suggestion # 1) for achieving minimum delays or to remove general phrases about efficiency from the paper.
  3. The description of the simulation model is still insufficiently detailed and clear. I suggest presenting the model parameters in tabular form for each experiment. Apparently, the simulation model used the average values of the parameters Ni. Using averages instead of random values reduces the value of the results. I suggest substantiating in more detail the sufficiency of using deterministic parameters of the simulation model instead of random ones.
  4. I suggest describing in more detail in the introduction or in the discussion the difference between the results obtained and similar studies, for example: https://doi.org/10.1155/2018/4702517, https://doi.org/10.1142/S0129183109014096 and the like, directly related to the subject of research.
  5. The question of the universality of the obtained results and dependencies remains open.
  6. The variable Qdi is used in formula 1, while in the explanation to the formula - Qi.

Author Response

RESPONSE TO COMMENTS OF REVIEWER №3

Article: “Influence of the parameters of the bus lane and the bus stop on the time delay of movement on private and public transport”

Fadyushin Alexey, Zakharov Dmitrii

The authors considered several suggestions and made the appropriate changes in the article. However, the authors considered the main proposals rather superficially.

Unfortunately, I am forced to repeat and concretize the main suggestions for improving the paper.

Point 1:

The authors have removed the words "optimization" and "efficiency" from the title of the paper. However, no corresponding changes were made in the article itself. The results of the study do not allow us to explicitly determine the optimal parameters of the bus lane and the bus stop. Then is not clear what the phrase “The established dependencies will specify measures to optimize the parameters of the public transport infrastructure” means (lines 385, 386). Judging by the obtained dependencies, the delay time decreases without limitation as it increases, for example, removal distance of the bus stop. It is unclear simultaneously what values of the investigated parameters will be optimal. I suggest describing in detail the methodology for calculating the optimal values of the bus lane and the bus stop parameters based on the obtained dependencies.

Response 1:

The note has been removed. Information has been added and the research results are presented in the form of a mathematical model (equation 7, 14) and graphs (figure 5c, 9cd), showing the presence of quadratic dependences of values and optimal values at which the total delay time is minimal.

Point 2:

About efficiency. As you know, efficiency is an assessment of the result obtained, considering the costs incurred. When the authors say “The aim of the work is to improve the efficiency of traffic management ...” (line 58), they obviously mean only the result in the form of an increase in average speed (line 137), uniform loading of the road lanes (line 393), reducing the delay time. Simultaneously, the paper does not say anything about the costs to obtain this effect or restrictions. I suggest describing in detail the costs (restrictions - see suggestion No 1) for achieving minimum delays or to remove general phrases about efficiency from the paper.

Response 2:

The authors agree with the reviewer's observation that efficiency is the ratio of the result obtained to the cost incurred. The choice of the term “efficiency” was determined by the established definition adopted by the Federal Law 443 “Traffic Management”: the efficiency of traffic management is the ratio of time losses (delays) when vehicles and (or) pedestrians move before and after the implementation of measures to organize traffic during road safety conditions.

The note has been eliminated, the purpose of the work has been adjusted: the aim of the work is to reduce the delay time for movement by choosing the optimal parameters of the road infrastructure based on the established patterns of the influence of the parameters of the bus lane and bus stop on the delay time in movement (rows 75-76).

Point 3:

The description of the simulation model is still insufficiently detailed and clear. I suggest presenting the model parameters in tabular form for each experiment. Apparently, the simulation model used the average values of the parameters Ni. Using averages instead of random values reduces the value of the results. I suggest substantiating in more detail the sufficiency of using deterministic parameters of the simulation model instead of random ones.

Response 3:

In PTV Vissim, simulation is performed with stochastic characteristics of vehicle inputs. To improve the accuracy of the simulation results, it is recommended to carry out several simulations under the same conditions. Preliminary experiments have shown that at LoS>1, the change in traffic parameters in several simulations is insignificant, so the results were calculated from one simulation.

The collection of results starts from 1200 seconds of simulation and continues for 3600 seconds. In a period of 0-1200 seconds, the road network model is filled with vehicles. At LoS>1, at the moment of the start of determining the traffic parameters (from 1200 seconds), traffic jams are formed in the simulation model with a traffic light object. And the stochastic nature of vehicle inputs does not affect the delay time. This makes it possible to achieve the correspondence of the parameters of the simulation model to the real traffic conditions.

The authors think that it is impractical to create such a table, since the table will contain 391 rows. For example, the factors L1.11 and NPrT in the model for experiments with L1.11 changed:

NPrT, veh./h

L1.11, m

20

40

60

80

500

Td

Td

Td

Td

1000

Td

Td

Td

Td

1500

Td

Td

Td

Td

2000

Td

Td

Td

Td

2500

Td

Td

Td

Td

The third factor NPuT varied in the range of 60-240 Veh / h. Experiment matrices were created for 3 simulation models (model for experiments with L1.11, model for experiments with L1.5 and model for experiments with bus stop parameters).

We think that such figures of the simulation model are not required for presentation, since the simulation model has a standard design and schematic representations of the models are presented in the article.

At each point, 6 ten-minute segments were simulated, and the article presents the average value of these parameters.

Point 4:

I suggest describing in more detail in the introduction or in the discussion the difference between the results obtained and similar studies, for example:

https://doi.org/10.1155/2018/4702517,

https://doi.org/10.1142/S0129183109014096

and the like, directly related to the subject of research.

Response 4:

We agree with the remarks, the analysis of previously performed studies related to the location of the configuration and parameters of stopping points (rows 46-68, references [21, 22, 26-35]) has been carried out in more detail.

Point 5:

The question of the universality of the obtained results and dependencies remains open.

Response 5:

The authors believe that the results and dependencies obtained are universal and can be applied to other transport systems of cities, for the following reasons:

- technical characteristics of vehicles (Car, Bus, Truck) operated in the USA, EC, Russia are approximately at the same level;

- the rules of the road adopted in different countries of the world are often like, since these countries (including Russia) have joined the Vienna Convention on Road Traffic;

- traffic organization schemes (including Bus lane and Bus stop) are approximately the same in different countries;

- the versatility of software products for traffic modeling;

- the problem of traffic jams and improving the level of service is typical for large cities in all countries of the world.

Point 6:

The variable Qdi is used in formula 1, while in the explanation to the formula - Qi.

Response 6:

Index “d” in the formula 1 removed. It's a typographical error.

Round 3

Reviewer 3 Report

The authors responded in sufficient detail to the reviewer’s suggestions. Unfortunately, some of these answers have not yet been accurately reflected in the paper itself and are not available to future readers.

I suggest presenting in the paper a detailed description of the following points.

  1. A detailed description of the experimental conditions should be clear to the readers. Readers should clearly understand under what conditions (limitations) the experimental results were obtained. I suggest including in the paper the author's answer to my suggestion No. 3 from the previous review. This answer in the paper should be presented in an academic style.
  2. An assessment of the universality of the results obtained (the authors' response to suggestion No. 5 of my previous review) should also be presented in the paper.
  3. The question of the optimal parameters of the infrastructure remains open.
    • It is not clear how equations 7 and 14 were obtained. If these are known equations, then reference to the original research is necessary.
    • How is "minimum value Td, m." in formulas 7 and 14? Do I understand correctly that these equations allow you to calculate the values of the delay time at the already known minimum values of this time? How correct is this?
    • I suggest presenting an empirical formula for the graphs (Fig. 5c and 9c, d) by analogy with the graphs of linear functions.
    • The specific numerical values of the optimal parameters of the infrastructure for the simulated conditions and constraints remained unclear. I suggest presenting specific numerical values of the optimal parameters of the infrastructure, since the presented graphs and functional dependencies do not give a clear answer to this question.

Dear Authors please include your explanations and responses to the reviewers' suggestions directly in the paper. Do it in an academic style. All questions and suggestions from reviewers are aimed at increasing the readability and interest of readers in your paper.

Author Response

Point 1:

A detailed description of the experimental conditions should be clear to the readers. Readers should clearly understand under what conditions (limitations) the experimental results were obtained. I suggest including in the paper the author's answer to my suggestion No. 3 from the previous review. This answer in the paper should be presented in an academic style.

Response 1:

The authors agree with the reviewer, the response to suggestion No. 3 (of the previous review) has been added to the article (lines 123-133, 194-200).

Point 2:

An assessment of the universality of the results obtained (the authors' response to suggestion No. 5 of my previous review) should also be presented in the paper.

Response 2:

The authors agree with the reviewer, the response to suggestion No. 5 (of the previous review) has been added to the article (lines 435-445).

Point 3:

The question of the optimal parameters of the infrastructure remains open.

It is not clear how equations 7 and 14 were obtained. If these are known equations, then reference to the original research is necessary.

How is "minimum value Td, m." in formulas 7 and 14? Do I understand correctly that these equations allow you to calculate the values of the delay time at the already known minimum values of this time? How correct is this?

I suggest presenting an empirical formula for the graphs (Fig. 5c and 9c, d) by analogy with the graphs of linear functions.

The specific numerical values of the optimal parameters of the infrastructure for the simulated conditions and constraints remained unclear. I suggest presenting specific numerical values of the optimal parameters of the infrastructure, since the presented graphs and functional dependencies do not give a clear answer to this question.

Response 3:

Equations 7 and 14 were obtained in our study. In the third edition of the article, the authors decided to change these formulas.

“Minimum Td value” is a typographical error when decoding the Td0 indicator, meaning not minimal, but optimal. Tdi0 is the total delay time at the optimal length of the lane marking 1.11 or 1.5. When the value of L1.11 or L1.5 deviates from the optimal value, TdPrT increases.

Empirical formulas have been added to the figures.

The note has been eliminated; graphs and figures have been added to the article (fig. 7, 11), the content of the article has been changed describing the determination of the optimal parameters of the infrastructure.
